# Error Reduction in Vision-Based Multirotor Landing System

**DOI:** 10.3390/s22103625

**Published:** 2022-05-10

**Authors:** Juan Pedro Llerena Caña, Jesús García Herrero, José Manuel Molina López

**Affiliations:** Applied Artificial Intelligence Group (GIAA), Carlos III University of Madrid, 28270 Madrid, Spain; molina@ia.uc3m.es

**Keywords:** UAV, autonomous landing, filtering, computer vision, helipad context, global position, navigation system, SITL

## Abstract

New applications are continuously appearing with drones as protagonists, but all of them share an essential critical maneuver—landing. New application requirements have led the study of novel landing strategies, in which vision systems have played and continue to play a key role. Generally, the new applications use the control and navigation systems embedded in the aircraft. However, the internal dynamics of these systems, initially focused on other tasks such as the smoothing trajectories between different waypoints, can trigger undesired behaviors. In this paper, we propose a landing system based on monocular vision and navigation information to estimate the helipad global position. In addition, the global estimation system includes a position error correction module by cylinder space transformation and a filtering system with a sliding window. To conclude, the landing system is evaluated with three quality metrics, showing how the proposed correction system together with stationary filtering improves the raw landing system.

## 1. Introduction

The growing demand for drone applications motivates the study of the support technologies for this type of small and powerful unmanned aerial vehicle (UAV). However, all new applications share an essential and critical maneuver—landing.

Generally, work in the literature about landing maneuvers, both for fixed and rotary wing UAVs, focuses on control strategies [1,2,3]. All of them require access to the internal vehicle states, the actuators or specific modes of the control or navigation system.

Under the precision landing concept is included all the solutions that approach this maneuver in an autonomous or supervised way, independent of the techniques and sensors used to estimate the vehicle states such as position or orientation, as well as its corresponding velocities and accelerations. The landing maneuver can be included within the navigation system where two main groups can be distinguished, outdoor and indoor navigation. Generally, outdoor navigation is based on the Global Navigation Satellite System (GNSS), but practically all current systems use fusion techniques that allow the integration of different strategies for estimating one or more vehicles’ states, which is necessary for the control and/or navigation system. Some of the most common cases in navigation systems are the use of barometers and/or sonars to improve the accuracy of the altitude provided by GNSS, or the use of small zenith cameras to determine small horizontal displacements [4]. Other specific navigation techniques such as visual odometry or visual Simultaneous Location And Mapping (SLAM) [5] are beginning to be used in indoor navigation.

Thus, other landing works focus on improving the accuracy of instrument systems such as [6,7,8] or even context information such as safe landing zones, as in work by Shah Alam, Md et al. shown in work [9]. Some commercial solutions focus on the use of beacons that indicate the landing region, as can be seen in the work of J. Janousek and P. Marcon [7] using a commercial infrared light beacon. This type of system includes an external controller to minimize the pixel error between the region of interest (ROI), defined by the centroid of the infrared area and the reference in the image plane, generally located at the center of the image plane. These strategies need to access internal vehicle states such as velocity or acceleration to correct the error.

In terms of context information, vision systems proved to be efficient to identify ROIs. In addition, knowing the landing context, specifically where or how the helipad is where the UAV must land, can help to improve the landing maneuver.

Developing strategies to identify and understand the context information of the aircraft allows providing the systems with greater autonomy. In the survey of autonomous landing techniques for UAVs by Alvika Gautam et al. [1], the authors describe the relationship between sensors/navigation systems and aircraft control modules, paying particular attention to vision landing techniques, generally responsible for recognizing and estimate the helipad position.

Some civil and commercial UAVs, such as certain DJI models [10], are beginning to integrate vision-based precision landing systems. In the work of Yoakum and Carreta [11], the authors conduct a study of the precision landing system of a DJI Mavic Pro, proving the aircraft and the integrated landing system meet specific accuracy requirements to use a specific wireless charging station.

Generally, vision systems for landings focus on identifying the landing area, either by means of context information of the helipad pattern or by the terrain conditions. The work of Mittal et al. [12] is an example of the identification of landing area conditions, where terrain slope is estimated to verify the feasibility of a UAV to land in urban search and rescue.

Regarding pattern recognition landing systems, works such as [13,14], among others [15,16,17,18], focus on finding the position using known patterns by Perspective-n-Point (PnP) algorithms [19]. Patterns such as Aruco [20], charuco, or new fractal patterns such as [21] or the deep learning trend You Only Look Once (YOLO) [22] try to improve the pattern pose estimation and prove to be widespread systems in the literature. In the literature and throughout this paper, an object “pose” means a set of position and heading of a specific reference system.

On the other hand, the emergence of open-source flight controllers such as Pixhawk [23], together with specific communication protocols such as Micro Air Vehicle Link (MAVLink) [24] and multiplatform APIs such as MavSDK [25], help to develop new applications and research new landing strategies.

PX4 [26] autopilot set up as a rotary wing vehicle has a planification system that allows dynamically smoothing the trajectories between different positions [27]. Specifically, it smooths the trajectories between consecutive waypoints by rounding the turns with radii over the waypoints and increasing or decreasing the drone speed when approaching or moving away from a waypoint [28,29,30]. The guidance algorithm integrated in PX4 is the 
$L_{1}$ algorithm introduced by Park et al. [31] under the linear approach. When forcing a new target position while the vehicle is navigating between two positions, the system changes target and tends to reach the newly added location by smoothing its current trajectory, as shown in works such as Stateczny et al. [32]. This behavior, when repeated with a certain frequency to include new waypoints referring to the same position, but with a certain noise, produces a spin effect on the aircraft that we call “inter-waypoint noise spin effect”.

In this work, we propose using the aircraft guidance system without downing the controller level for a widespread implementation of the precision landing system, contributing to UAV air safety and helping the emergence of new applications.

We propose a new contribution with respect to classical geolocation landing strategies based on global positioning by decoupling the landing in two phases, first reaching the target coordinates and then activating the landing mode, descending vertically with a constant descent speed α.

Our proposed landing strategy seeks to descend quickly when the target is found and to smooth its descent as the aircraft approaches, without having to adjust the controller parameters. This idea attempts to improve the image resolution quickly to improve the position estimation. The difference with respect to other works is that simultaneously descending and adjusting the positioning relies on the variable waypoint altitude adjustment without accessing the controller, so that without changing the internal descent controller of the PX4 [26] or adding new external control laws, the strategy allows smoothing the descent when approaching the target. This strategy allows taking further steps in the final phase of the approximation, improving the final estimate, and ensuring the stability of the system provided by the manufacturer. For this, we propose a function to modulate the default behavior of the PX4 controller which seeks a stationary descent at a constant speed.

In addition, this paper models the error of a precision landing system using a monocular vision system, context information from the helipad pattern, and the internal navigation system of the PX4 flight controller.

The vision system error modeling allows a fine calibration of helipad localization. This correction, together with a stationary filtering of the estimates by sliding time window and variable adjustment of the descent height, allow to reduce the spin effect produced by the estimation error of a vision system when integrated with the 
$L_{1}$ navigation algorithm, without access to the internal controller parameters or relative states of the aircraft that may require re-programming of the on-board computer.

To sum up, this paper presents two main contributions in precision landing by vision-based global position: the continuous adjustment of the approach to descent trajectory, and the improvement of the position by vision through systematic error adjustment and filtering.

The proposal strategy is evaluated after including an error model correction as well as different sliding time window filters. The work is developed on a hyper-realistic software in the loop (SITL) [33] simulation system with the PX4 flight controller and the AirSim [34] simulator.

Finally, the results of the study show the estimation error analysis and filtering of the estimates with sliding time window filters, minimizing the inter-waypoint noise spin effect generated by noisy waypoint transitions and improving three quality metrics of landing time, trajectory landing length, and landing accuracy without additional control law, enabling the use of the aircraft’s guidance system as an alternative for the deployment of precision landing technology.

This paper is organized as follows: Section 2 shows the problem formulation of helipad spatial estimation by monocular computer vision. Section 3 describes the landing strategy proposal and the global estimation module. The correction module design, the analysis of the complete landing system, and a description of the test environment can be found in Section 4. Finally, the conclusions are presented in Section 5.

## 2. Problem Formulation

We consider the problem of a UAV landing on a certain landing pad using its internal autopilot waypoint guidance system and a monocular vision system with gimbal integrated in the UAV.

Figure 1 shows the set of reference frames, where the superscripts 
{t,ph,c,z,g,b,n,e and g} correspond to the reference frames of the helipad (target), pinhole camera model, camera, gimbal socket for the camera, gimbal, body, North-East-Down (NED), Earth-Centered Earth-Fixed (ECEF), and global. 

In this way, any given point 
$p_{t}^{t}\;$expressed in a flat pattern reference frame 
{t} and the homogeneous transformation 
${}^{n}T_{t}$ between the landing pad reference frame to the NED referent frame 
{n} can be expressed in the global reference frame 
{g} system 
$p_{t}^{g}$ applying a set of transformations shown in Equations (1) and (2).
(1)
$$p_{t}^{n}=\overset{{}^{n}T_{t}}{\overbrace{{}^{n}T_{b}\cdot {}^{b}T_{g}\cdot {}^{g}T_{z}\cdot {}^{z}T_{c}\cdot {}^{c}T_{t}}}\cdot p_{t}^{t}$$
(2)
$$p_{t}^{g}={}^{g}F_{e}\left\{{}^{e}F_{n}\left[p_{t}^{n},{}^{e}F_{g}(p_{Ref}^{g}\right)]\right\}$$
where superscript 
{j} over point 
$p_{i}^{j}$ means the reference frame system, and the subscript 
i={t,Ref} denotes the name of the point (target and reference). 
${}^{j}T_{i}$ means the homogeneous transformation between reference frame 
i and 
j. On the other hand, 
${}^{j}F_{i}$ refers to nonlinear transformations between reference frame 
i and 
j. 
$p_{Ref}^{g}\;$indicates the global position of the body (UAV) as a global reference point.

The set of reference frame systems involved in the transformations are shown in Figure 1 and denoted as: helipad (target) 
{t}, pinhole camera model 
{ph}, camera 
{c}, gimbal socket for the camera 
{z}, gimbal 
{g}, body 
{b}, North-East-Down (NED) 
{n}, Earth-Centered Earth-Fixed (ECEF) 
{e}, and global 
{g}.

### 2.1. Pattern (Helipad) Detection

We consider as the helipad a reference pattern defined by an Aruco pattern [35] with a certain number of bits, as part of a library 
B. As shown in the paper [20], the system identifies candidate square regions as Aruco markers, then encodes these regions and compares them to the pattern dictionary as desired.

The full process can be divided into the following steps:**Image conversion:** Obtain an RGB image and transform it to grayscale.**Edge extraction**: We understand as edge an intensity change boundary, some classical algorithms are Canny [36] and Sobel [37].**Contour extraction**: We understand a contour as a curve of points without gaps or jumps. Therefore, the objective is to identify if the edges found represent contours. An example of simple contour extraction can be given by a binarized image of an object whose outer contour can be extracted by subtracting the original binarized-dilated image from the original binarized image. To check if closed regions appear, a segmentation by connected components would provide us candidate regions of interest (ROI) as a result.**Contour filtering**: Only show rectangular regions.**Removing ROI perspective distortion**: For this it is necessary to find the general plane 
$P^{2}$ projective transformation 
$h:P^{2}\to P^{2}|\;h\left(m\right)=m^{\prime }=mH$, where 
m is a point in a plane. 
$H^{3\times 3}\;$is a non-singular matrix where 
m′ is the linear transformation 
H of 
m. The transformation 
H is biunivocal and homogeneous, in other words, a point over a plane is a unique point over another plane and 
kH| k∈ℝ and 
k≠0 is also the solution. This condition allows dividing the matrix 
H by the element 
$h_{33},$ decreasing the dimension of terms to identify from 9 to 8. The correspondence between points 
$\left(x_{i},y_{i}\right)↔\left(x_{i}^{\prime },y_{i}^{\prime }\right)$ can be expressed in matrix form as 
$b_{i}=A_{i}h,$ and their relationship is expressed as Equation (3) (more details in [38]). Knowing 
n pairs of points, the system of 
2n equations and 8 unknowns is established as 
b=Ah, where 
$A={\left[A_{1},A_{2},\ldots ,A_{n}\right]}^{T},\;b={\left[b_{1},b_{2},\ldots ,b_{n}\right]}^{T},\;$and 
$h^{3\times 3}$ matrix as 
$h_{33}=1$. For 
n=4, the direct solution 
$h=A^{-1}b$; if 
n>4 the system is overdetermined and least squares can be applied, 
$h={\left[A^{T}A\right]}^{-1}A^{T}b$. For cases where 
$h_{33}=0$ refer to [38].
(3)
$$x^{\prime }=\frac{h_{11}x+h_{12}y+h_{13}}{h_{31}x+h_{32}y+h_{33}}\;y^{\prime }=\frac{h_{21}x+h_{22}y+h_{23}}{h_{31}x+h_{32}y+h_{33}}\;A=\left[\begin{array}{c} \begin{array}{cc} x & \begin{array}{ccc} \begin{array}{cc} y & 1 \end{array} & \begin{array}{cc} 0 & 0 \end{array} & \begin{array}{ccc} 0 & -x\prime x & -x\prime y \end{array} \end{array} \end{array}\\ \begin{array}{ccc} \begin{array}{cc} 0 & 0 \end{array} & 0 & \begin{array}{cc} x & \begin{array}{cc} y & \begin{array}{ccc} 1 & -y\prime x & -y\prime y \end{array} \end{array} \end{array} \end{array} \end{array}\right];b=\left[\begin{array}{c} x^{\prime }\\ y^{\prime } \end{array}\right];h={\left[h_{11},h_{12},h_{13},h_{21},\ldots h_{32}\right]}^{T}$$**Pattern library matching check**: The binary code of the ROI is extracted, superimposing on the binarized and perspective-corrected image a grid of the same cell size as the searched one. Each grid cell receives a binary value according to if the corresponding color is black (zero) or white (one). The Hamming coding algorithm (ref) is applied to the extracted code to eliminate false negatives. This resulting code is compared with the selected pattern dictionary, filtering the regions identified as markers and belonging to the pattern dictionary from other regions. In addition, this step provides information about the marker id if the ROI belongs to the library.

### 2.2. Helipad Pose Estimation

For pose estimation, the Perspective-n-Point (PnP) problem [35] is formulated where the objective is to minimize the reprojection error Equation (7) of 3D points in the image plane 
{ph}. This problem is closely linked to a calibrated system, since it requires a camera model, pinhole, and a pattern that allows to relate identified features of an image with features of the pattern.

Given a point 
ptc∈ ℝ3 belonging to the knowing pattern located in real-world 3D space and expressed in the camera reference frame 
{c}, it can be expressed in the image camera plane reference frame 
{ph} as 
ptph∈ℝ2. The relationship between the two reference frames is provided by the pinhole camera model in Equation (4).
(4)
$$sp_{t}^{ph}=Ap_{t}^{c}$$
where 
s is a scale factor and 
A intrinsic camera matrix [17]. The internal matrix 
A is composed of the focal distances 
$\left(f_{x},f_{y}\right)$ and the principal points 
$\left(c_{x},c_{y}\right)$. The pinhole model can be improved with radial, tangential, or prism distortion corrections, adding 
n set of 
$k_{i}$ parameters to the model [39,40,41]. The set of internal parameters of the camera model can be expressed by the vector 
$\delta =\left(f_{x},f_{y},c_{x},c_{y},\;k_{1},\;\ldots ,\;k_{n}\right)$.

If the point is expressed in coordinates of the pattern reference frame 
$p_{t}^{t}$, there exists an extrinsic homogenous transformation 
${}^{c}T_{t}$ to relate the reference frame of the pattern to the camera reference frame Equation (5) is used.
(5)
$$p_{t}^{c}={}^{c}T_{t}p_{t}^{t}$$
where the transformation 
cTt=[cRt|vtc] is a rototranslation composed of the pattern’s orientation 
cRt=Rx(θ1)Ry(θ2)Rz(θ3) to the camera and the pattern position vector to the camera 
vtc∈ℝ3. Thus, the parameter vector to be identified to obtain the camera–pattern relationship is 
θ=(R,v)=(θ1,θ2,… ,θ6)∈ℝ6.

Joining Equations (4) and (5) and adding distortion models, the camera model remains as a Function (6) that projects points 
ptt∈ℝ3 to 
ptph∈ℝ2 points of the camera image plane.
(6)
$$p_{t}^{ph}=\Psi \left(\delta ,\theta ,p_{t}^{t}\right)$$

Then, helipad pose estimation is the problem of minimizing the reprojection error Equation (7) of the observed helipad pattern features. One of the classic features to identify by computer vision are the corners. If the pattern is known, we know a priori the 3D position of these corners in the reference frame of the pattern.
(7)
$$\hat{E}=arg\;min\underset{p_{i}^{t}\in ∁}{\sum }{\left[\Psi \left(\delta ,\theta ,p_{i}^{t}\right)-O\left(p_{i}^{t}\right)\right]}^{2}\;$$
where 
$p_{i}^{t}\in ∁$ and 
∁ is a corner set of the pattern. 
O(pit)∈ℝ2 is the corners obtained in the camera plane by a specific computer vision algorithm such as the Harris or Susan algorithm [22,42].

Furthermore, since all points 
$p_{i}^{t}$ belong to a pattern plane, the 
z-component of all corners in the pattern frame will always be 0. This quality allows solving Equation (7) using specific methods such as the Infinitesimal Plane-Based Pose Estimation (IPPE) [43].

The estimation of internal camera parameters requires a learning phase modeled in Equation (7) as an optimization problem. In addition, identifying the six parameters to define the transformation 
${}^{c}T_{t}$ between the pattern calibration and the camera involves a similar process. Although both processes can be clustered as shown in Equation (7), the internal camera parameters 
δ will be constant for a particular vision system; however, the position of the pattern may change. For this reason, it is decoupled in two phases: on the one hand, a camera parameter learning (calibration) process, using a set of images of a known pattern to estimate internal camera parameters, and, on the other hand, the estimation of the helipad position for a certain image during flight.

### 2.3. Camera-Gimbal Frame

The camera is placed in a camera-gimbal socket, so it is necessary to include this referent frame 
{z}. As the camera-gimbal socket axis is equivalent with the general gimbal axes, but static, the 
${}^{z}T_{c}$ transformation is shown in Equation (8).
(8)
$${}^{z}T_{c}=\left[R_{c}^{z}|0^{3\times 1}\right]\equiv \left(\begin{array}{cc} R_{c}^{z} & 0^{3\times 1}\\ 0^{1\times 3} & 1 \end{array}\right);\;R_{c}^{z}=R\left(x,\frac{\pi }{2}\right)R\left(z,\frac{\pi }{2}\right)$$
where 
R(x,θ) and 
R(z,ψ) represent 
θ and 
ψ rotations about the 
x and 
z axes of the camera reference system. As the axes of the gimbal and camera-gimbal socket are equivalent, the relationship between camera-gimbal socket and gimbal corresponds to the identity matrix 
${}^{g}T_{z}=I^{4\times 4}$.

### 2.4. Gimbal Body Frame

The gimbal’s reference frame 
{g} to the UAV’s body gravity center frame is defined as the composition of a roto-translation in Equation 
(9).
(9)
$${}^{b}T_{g}=\left[R_{g}^{b}|p_{g}^{b}\right]\;R_{g}^{b}=R_{g}\left(\theta ,\;\phi ,\psi \right);p_{g}^{b}={\left(x_{g},y_{g},z_{g}\right)}^{T}$$

For this work, we consider the gimbal is static but located at the 
$p_{g}^{b}$ position with 
$R_{g}\left(\theta ,\;\phi ,\psi \right)$ rotation to the body axes.

In this paper we consider 
${}^{b}T_{z}=\left[R_{b}\left(0,-\frac{\pi }{2},0\right)|{\left(0,0,0.1\right)}^{T}\right],$ as shown in Section 4.1.

### 2.5. Body-NED Frame

The North-East-Down (NED) frame coordinates 
{n} to the UAV body gravity center 
{b}, 
${}^{n}T_{b}$ is equivalent to the body rotation at the angle defined by the yaw angle 
ψ to geographic north or azimuth, pitch attitude to horizon plane 
ϕ, and roll angle defined to gravity direction 
θ. These angles refer to the attitude and heading reference system (AHRS) frame of reference that groups magnetic, angular rate, and gravity (MARG) information. Generally, these systems usually include air data to provide altitude or wind speed information.
(10)
$${}^{n}T_{b}=[R_{b}^{n}|0^{3\times 1}];\;R_{b}^{n}=\left(\begin{array}{ccc} \cos\theta \cos\psi & \sin\psi \sin\theta \sin\phi -\sin\psi \cos\phi & \sin\psi \sin\theta \cos\phi +\cos\psi \sin\phi \\ \cos\theta \sin\psi & \cos\psi \cos\theta +\sin\psi \sin\theta \sin\phi & \sin\psi \sin\theta \cos\phi -\cos\psi \sin\phi \\ -\sin\theta & \cos\theta \sin\phi & \cos\theta \cos\phi \; \end{array}\right)$$

### 2.6. NED-ECEF-Global

The coordinate transformation between NED to the global reference frame 
{g} requires the use of the Earth-Centered Earth-Fixed (ECEF) reference system 
{e}, which allows us to apply the corresponding geodetic transformations to the terrestrial model and finally obtain the coordinates in global terms. In our case, we use a WGS84 (World Geodetic System 84) [44] datum.

The constant parameters of the WGS84 datum in Figure 2 refer to: 
$r_{e}$ semimajor axis (equatorial radius), 
$r_{p}$ semiminor axis (polar axis radius), 
ε first eccentricity and 
ε′ second eccentricity of the ellipsoid. It is important to differentiate the geocentric coordinates, referred to as the ECEF system, from the geodetic coordinates, referred to as the geodetic model (WGS84). This difference is provided by the geodetic model (datum) and is represented in the diagram on the right of Figure 2, where 
$\varphi^{\prime }$ refers to geocentric latitude and 
φ refers to geodetic latitude.

Given a point 
$p_{t}^{n}$ expressed in NED reference frame 
{n} of a local tangent plane (LTP) to a geodesic surface at a known point 
$p_{Ref}^{g}={\left(\lambda ,\varphi ,h\right)}^{T}{}_{Ref}$, it can be expressed in ECEF coordinates 
{e} applying Equation (12). This equation corresponds to a translation in ECEF reference frame. However, to obtain 
$p_{Ref}^{e}$ coordinates of our reference point in ECEF frame it is necessary to transform the global coordinates to ECEF applying Equation (14). The transformation between local coordinates and ECEF is given by the transformation Equation (13). In this work, we consider 
$p_{Ref}^{g}=p_{UAV}^{g}$.

On the other hand, a given point 
$p_{t}^{e}$ expressed in ECEF can be expressed in global coordinates 
$p_{t}^{g}$ applying the transformation Equation (11).
(11)
$$p_{t}^{g}={\left(\begin{array}{c} \lambda \\ \varphi \\ h \end{array}\right)}_{t}={}^{g}F_{e}\left(p_{t}^{e}\right)=\left(\begin{array}{c} {tan}^{-1}\;\left(\frac{y_{t}^{e}}{x_{t}^{e}}\right)\\ {tan}^{-1}\left(\frac{z_{t}^{e}+e^{\prime 2}Z_{0}}{r}\right)\\ U\left(1-\frac{r_{p}^{2}}{r_{e}V}\right) \end{array}\right)$$
where 
${\left(\lambda ,\varphi ,h\right)}^{T}$ means longitude, latitude, and altitude in WGS84 datum. 
${\left(x_{t}^{e},y_{t}^{e},z_{t}^{e}\right)}^{T}$ are the coordinates in the ECEF reference frame. The transformation Equation (11) corresponds to Jijie Zhu’s algorithm [45] analyzed and compared in [46].
(12)
$$p_{t}^{e}=\left(\begin{array}{c} x_{t}^{e}\\ y_{t}^{e}\\ z_{t}^{e} \end{array}\right)={}^{e}F_{n}\left(p_{t}^{n},p_{Ref}^{e}\right)=R_{n}^{e}\cdot p_{t}^{n}+p_{Ref}^{e}$$
(13)
$$R_{n}^{e}=\left(\begin{array}{ccc} -sin\varphi_{Ref}cos\lambda_{Ref} & -sin\lambda_{Ref} & -cos\varphi_{Ref}cos\lambda_{Ref}\\ -sin\varphi_{Ref}sin\lambda_{Ref} & cos\lambda_{Ref} & -cos\varphi_{Ref}cos\lambda_{Ref}\\ cos\varphi_{Ref} & 0 & -sin\varphi_{Ref} \end{array}\right)$$
(14)
$$p_{Ref}^{e}=\left(\begin{array}{c} x_{Ref}^{e}\\ y_{Ref}^{e}\\ z_{Ref}^{e} \end{array}\right)=\left(\begin{array}{c} \left(r_{\lambda }+h_{Ref}\;\right)cos\varphi_{Ref}cos\lambda_{Ref}\\ \left(r_{\lambda }+h_{Ref}\right)cos\varphi_{Ref}sin\lambda_{Ref}\\ \left(\left(1-\varepsilon^{2}\right)r_{\lambda }+h_{Ref}\right)sin\varphi_{Ref} \end{array}\right)\;$$
(15)
$$r_{\lambda }=\frac{r_{e}}{\sqrt{1-\varepsilon^{2}\sin^{2}\varphi \;}}$$

## 3. Proposal

In this section, first the landing strategy is described, then the method to determine the helipad’s global position is detailed, and finally the error analysis of the helipad’s position estimation is given.

### 3.1. Landing Strategy

The landing strategy is responsible for telling the UAV navigation system the position to which it must go and the attitude it must have to align with the target (Algorithm 1). The position of the target is static, but the attitude and altitude to helipad vary overtime when the UAV attempts to land.
**Algorithm** **1** Landing Strategy1:
$\left[p_{t}^{g},{\left(\theta ,\phi ,\psi \right)}_{t}^{g},a\right]=\mathrm{helipad}\;\mathrm{identification}\;h_{UAv},\psi_{UAV}=\mathrm{UAV}\;\mathrm{navigation}\;\mathrm{sistem}\;$2:
if a=True3:Buffer 
← [
$p,\;\left(\theta ,\phi ,\psi \right){]}_{t}^{g}$4:
if frequency=1Hz & Buffer≥10 5:
$p_{t}^{\prime g},\;\psi_{t}^{\prime g}=Filter\;\left(Buffer\right)$6:Buffer reset 7:Buffer (1) 
$\leftarrow {\left[p^{\prime },\;\;\psi^{\prime }\right]}_{t}^{g}$8:
$\psi_{set}=\psi_{UAV}^{n}+\psi_{t}^{b}$9:
${\left[\lambda ,\varphi \right]}_{set}\leftarrow p_{t}^{\prime g}$10:
$h_{set}=h_{UAV}\left(1-0.1e^{\frac{-1}{h_{t}^{b}-0.5h_{0}}}\right)$11:UAV navigation planer 
← [
$\lambda_{set},\;\varphi_{set},h_{set},\;\psi_{set}]$12:
$if\;h_{UAV}\leq h_{0}$13:PX4 landing mode14:break15:**end if**16:**else**17:
goto→1 18:**end if**19:**else**20:
goto→121:**end if**

#### 3.1.1. Helipad Azimuth

To align the drone to the marker, it is necessary to determine the azimuth of the marker 
$\psi_{t}^{n}$. For this, we use the azimuth of the drone 
$\psi_{UAV}^{n}\;$and the orientation of the marker to the drone 
$\psi_{t}^{b}$.

Figure 3 shows how the helipad azimuth can be obtained graphically by adding to the drone azimuth the orientation of the aircraft to the landing pad in Equation (16). When both systems are aligned, the marker azimuth will be equal to the drone azimuth.
(16)
$$\psi_{t}^{n}=\psi_{UAV}^{n}+\psi_{t}^{b}$$

#### 3.1.2. Altitude Setpoint Strategy

In order to change the default controller descent behavior, it is possible to use the behavior of the system in the transient state, i.e., before reaching the maximum velocity of stationary descent. Thus, if the new desired height is reached without having to reach the maximum descent speed, the behavior will be smooth, and if the destination point is far enough away, the controller saturates and descends with maximum constant speed behavior, without exceeding the internal controller parameters.

To define step points that allow a linear descent at constant speed α in an iterative loop, the new step point will correspond to the current height minus a certain parameter α.
(17)
$$h_{set}=h_{UAV}-\alpha$$

Considering this process is iterative (discrete) with a sample time of 
∆t, the previous equation can be expressed as follows:(18)
$$h_{t+1}=h_{t}-\alpha ∆t$$
where 
t subscript means instant time. Solving the 
α term, it is verified that alpha corresponds to a speed term.
(19)
$$\frac{\left(h_{k+1}-h_{k}\right)}{∆t}=\alpha =\frac{∆h}{∆t}=cte.\;$$

In our case, the aim is to design the 
$h_{set}\left(h_{UAV}\right)$ function such that the aircraft approaches with a smooth behavior to 
$h_{0}\;$and at that point lands automatically with internal autopilot.

To perform this, we propose Function (23).
(20)
$$h_{set}=h_{UAV}\left(1-\beta_{1}e^{\frac{-1}{h_{t}^{b}-\beta_{0}h_{0}}}\right)$$
where
$\;\beta_{1}$ is the weight of the exponential function and 
$\beta_{0}<1,$ which allows slightly shifting the value of 
$h_{0}\;$and to be able to switch to automatic landing mode. The 
$\beta_{1}=0.1$ value is set heuristically, while 
$\beta_{0}=0.5$ is set to shift 50% less than the switching height 
$h_{0}$. The system must consider the relative flight altitude 
$h_{UAV}$ and the height of the UAV relative to the landing pad 
$h_{t}^{b}$.

Figure 4 shows an approximate representation of the altitude-set function behavior Equation (20) vs. constant decreasing Equation (17).

Figure 4 shows the approximate descent behavior of our proposal versus a constant speed descent. In the final phase of the approximation, the descent becomes smoother than in the linear behavior. The Figure 4 behavior should be taken as an illustrative example of the desired behavior, not as a realistic simulation. The final behavior can be seen in experimentation.

#### 3.1.3. Filter

The states to be filtered are the global position of the helipad 
$p_{t}^{g}=\left(\lambda_{t},\;\varphi_{t}\right)$ and its orientation to north or azimuth 
$\psi_{t}^{n}$. All these variables are static, since the landing pad is static; therefore, the filter model does not need to provide information for each new measurement, rather we need to know their stationary statistical values. For this we propose to generate a data buffer with memory. The size of the buffer defines the size of the filtering window 
L. The initialization saves 
L new measurements and then finds the mean or median of the buffered data. Finally, the buffer is reset.

To propagate the information over time, the sliding window does not overlap with previous values, but the value filtered at the previous instant is included as the first measurement in the clean buffer.

As for the filter memory, if the new values change substantially it will vanish in the long term, since the weight given to the past values is 
$\frac{1}{L}$ versus 
$\frac{L-1}{L}$ for each new data, so the window size can be critical for cases where the target is moving. Finally, the size of the buffer/window 
L is linked to time thanks to the 
1 Hz system sampling time to provide new measurements.

### 3.2. Helipad Global Position Estimation

The helipad global position estimation system is responsible for integrating the vision system, the heliport context information, the gimbal, and the UAV navigation states, to provide the landing strategy with the helipad global position. In addition, this includes a spatial correction system for the NED frame, which is the objective of study of this work.

Figure 5 shows the diagram of the estimation system which is formulated in Equation (2). The system works as follows:Aircraft global position 
$p_{UAV}^{g}$ and attitude 
${\left(\theta ,\phi ,\psi \right)}_{UAV}^{b}$ is requested by the PX4 flight controller via MAVLink protocol [24] supported by the MAVSDK API [25].The gimbal position 
$p_{g}^{b}\left(x,y,z\right)\;$and attitude 
${\left(\theta ,\phi ,\psi \right)}_{g}^{b}$ is requested by the AirSim simulation environment via UDP protocol described in Section 4.1. This information composes the 
${}^{b}T_{z}$ transform.The vision system receives 
I image of 
W×H size and 3 RGB channels. The image is received via UDP protocol from the simulation system. In addition, the vision system has as input the context information from the helipad, the library (
Lib) of the marker, the marker’s identification number (
Id∈Lib), and the real marker’s size (
MS) in meters. The library is characterized by the number of horizontal and vertical bits (squares) that form the geometry of the marker and the number of elements that make up the library. The vision system output provides a Boolean variable 
a∈B, that indicates if the landing pad has been detected or not. In addition, it provides the position of the landing pad to the camera 
$p_{t}^{c}$ and the attitude 
${\left(\theta ,\phi ,\psi \right)}_{t}^{c}$.The camera pose estimation (
$p_{c}^{t}$ and 
${}^{t}T_{c}$) is gated by the PnP method integrated in the OpenCV Aruco library [47] from a previously pre-calibrated camera (Test environment).The aircraft, gimbal, and landing pad position are combined in the set of 
${}^{n}T_{b},{}^{b}T_{z},$ and 
${}^{z}T_{c}$ transformations to obtain the positioning 
$p_{\;t}^{n}$ and attitude 
${\left(\theta ,\phi ,\psi \right)}_{t}^{n}$ of the landing pad in NED frame.The correction module provides the 
$p_{t}^{\prime n}$ positioning and attitude 
$\left(\theta ,\phi ,\psi \right)\prime_{t}^{\;n},$ tuned in NED coordinates.Finally, the target position in NED frame 
$p_{t}^{\prime n},$ together with the drone global position 
$p_{UAV}^{g}$ and ellipsoid WGS84 approximation, are used to obtain the helipad global position in Equation (11).

## 4. Landing System Analysis

The aim of this section is to evaluate the proposed estimation system and to identify the necessary corrections to be incorporated in the “correction” module of Figure 5. To achieve this, first the test environment and the necessary parameters are detailed in the subsection Test environment. Next, the system estimation error is modeled to provide the landing system a correction module. The quality of the correction is evaluated using the root mean square error (RMSE) together with the variation in the data distribution in terms of data distribution structure, mean, and standard deviation.

Finally, a full landing system and classical linear decreasing descent are compared and evaluated with four quality metrics, which quantify the trajectory length, the time to land, and the accuracy of landing on the helipad.

### 4.1. Test Environment

In this work, we use a hyper-realistic test environment based on Software in The Loop (SITL). SITL systems are simulation architectures where virtual world environments interact to simulate object, vehicle, and sensor together with external systems such as a flight controller or ground station, among others. These environments are powerful testing tools for earlier phases of system integration, as they allow realistic results to be obtained without potentially dangerous and expensive risks.

In our case, AirSim [48] is used as world environment and PX4 flight controller configured as a quadcopter. The simulated physical model corresponds to the Iris quadcopter and the set of sensors, and their specifications are detailed in Table 1. The models of the simulated sensors can be found detailed in [34].

Figure 6 shows an SITL communication diagram between the main system modules in SITL. The GCS module refers to the ground control station, in our case QGround control [50]. GCS is used to help to download the .log files generated in the test missions.

The vision-based estimation system requires knowledge of the internal camera parameters 
$\left\{A,k_{i}\right\}$. These parameters are obtained by standard calibration [39] using a chess pattern with nine rows, six columns and 20 cm sides of the squares. This pattern is integrated into the AirSim environment as a texture over a rectangular prism with 
1.8×1.2×1.8 [m] sides. To capture images, we implemented a system that automatically captures images while performing a spiral upward flight over the reference pattern. This allows obtaining a large set of images with the pattern from different positions.

Figure 7 shows the image of the calibration pattern in the AirSim reference frame, random image of the image registration process used for calibration, and a sample of the reprojection error.

Finally, the internal camera parameters are shown in Table 2. The context information used for the experimentation is: 
Lib=5×5×1000, Id=68, and 
MS=1 [m].

The landing system was developed in Python 3.6 with the AirSim [34] and MAVSDK [25] APIs. The experiments and the SITL environment were developed on a Windows Server 2019, 64 bits, hosted in AMD Ryzen 9 3900X 12-Core Processor CPU, 3.79 GHz with 64 GB RAM and 2×1TB SSD + 2×HDD 1.5TB of internal memory, graphic card Nvidia GeForce RTX 2060.

### 4.2. NED Error Modeling

To evaluate the estimation error, we propose to analyze the estimation data provided by the vision system over twenty static flights located at seventeen different positions at the same relative altitude above the ground, 10 meters.

The selected positions correspond to five different headings centered, 45° {northeast, southeast, southwest, and northwest} and four different distances 
$\left\{2,3,4,5\right\}\sqrt{2}$ to the takeoff origin where the helipad is located. In each position is recorded a total of 1000 
$p_{UAV}^{t}\;$samples. Looking at Figure 8, while the blue line maintains the desired directions of 45° (NE, SE, SO, NO), the centers of the positions recorded by the vision system, the red line, are decoupled, showing a constant angular deviation of the positions.

We consider the aircraft control system is asymptotically stable so that in steady state its position converges to the reference one. Thus, we consider as ground truth the reference positions for the steady flight.

The error position for each of the components is given by Equation (21).
(21)
$$e{\left(x\right)}_{i}^{f}=x_{i}^{f}-x_{GT_{i}}^{f}$$
where 
$e{\left(x\right)}_{i}^{f}$ means the position error of the component 
x of the system 
i in 
f referent frame. GT subscript means the ground truth in 
f reference frame.

Looking at the errors (Figure 9), the error distribution increases with increasing distance from the north-east plane origin (0, 0). This means the error position depends on the position in the NE plane.

Regarding altitude error, Figure 10a,b show for each twenty register positions a distribution with four “modes”. In Figure 10a these modes show as four scatter clusters and in Figure 10b as four peaks in each twenty distributions. In addition, the mean and median of the total error distribution, Figure 10, are displaced from the origin, showing a bias in altitude.

The different colors in Figure 10b show each of the twenty records, all of them showing four modes and centered on the same error terms. In this work, we focus on bias correction of mean and median; however, modeling the error in altitude is outside the scope of this paper.

The altitude error distribution may be a consequence of the internal discretizing of the simulator in the image render, so that the vision system, when segmenting the ROI of the helipad, extracts its contour with a size variation. This would be explainable according to the pinhole model of Equation (4) and PnP Formulation (7), since its scale factor is constant, but the size of the ROI and corner positions would change.

#### 4.2.1. Polar Space Error Analysis

The visual results in Figure 8 and Figure 9 show an apparent angular and radial bias of the helipad global position estimation system. We change the cartesian space to the cylindrical space defined by Equation (22), where the terms 
E,N,D are the coordinates in the NED referent frame.
(22)
$$r_{0}=\sqrt{E^{2}+N^{2}}\;\theta_{0}=atan\left(\frac{N}{E}\right)\;D_{0}=D$$

When plotting data in the new space in Figure 11, it can be seen how the data set is apparently clustered around a constant bias in the angle and radial error.

However, when showing the distance error (radial) behavior versus distance, Figure 12 shows a high linear correlation between distance and radial error. The angular error is also tested for linear dependence on distance, but the correlation does not exceed 
35% of variance score 
$r^{2}\;$in Equation (23), so it has been discarded.
(23)
$$r^{2}=1-\frac{Var\left(x-\hat{x}\right)}{Var\left(x\right)}$$
where 
$Var\left(x-\hat{x}\right)$ indicates the variance of the error between 
$\hat{x}$ model estimation and 
x data.

Figure 12 shows with “+” the radial error centers of each of the measurement positions and the linear model fitted (blue line) by least squares to these points. This model has a slope 
$\alpha_{r}=2.4923\times 10^{-2}$ and independent term 
$\beta_{r}=2.778497\times 10^{-2}\;\left[m\right]$. The linear model obtains around 94% of the variance score in Equation (23).

Given the results in Table 3 and Figure 12, the error bias in polar space can be tuned by the model of Equation (24), where the apostrophe over coordinates means corrected coordinate, subscript zero start value, and 
$\beta_{i}$ the bias of coordinate 
i.
(24)
$$r^{\prime }=r_{0}\left(1-\alpha_{r}\right)-\beta_{r}\;\theta^{\prime }=\theta_{0}-\beta_{\theta }\;D^{\prime }=D_{0}-\beta_{D}$$

#### 4.2.2. Error Correction in NED Space

Given 
N, 
E, and 
D coordinates of a point 
$p_{i}^{n}$ in the NED frame and knowing the correction in a cylindrical space in Equation (24), the objective is to return to the NED space. For this purpose, we apply the transformation Equation (25).
(25)
$$\hat{N}=\hat{r}.cos\left(\theta \right)\;\hat{E}=\hat{r}.sin\left(\theta \right)\;\hat{D}=D_{0}-\beta_{D}$$
where its terms 
$\hat{r},\;\theta \;$are taken as shown in Equation (26):(26)
$$\hat{r}=\left|v_{\beta_{\theta }}\right|-r^{\prime }\;r^{\prime }=\alpha_{r}r_{0}+\beta_{r}\;\theta =atan\left(\frac{v_{N}}{v_{E}}\right)\;v\left(\beta_{\theta }\right)=\{\begin{array}{c} v_{E}=E\cdot \cos\left(\beta_{\theta }\right)-N\cdot \sin\left(\beta_{\theta }\right)\\ v_{N}=E\cdot \sin\left(\beta_{\theta }\right)+N\cdot \cos\left(\beta_{\theta }\right) \end{array}$$
where the hat over 
N, 
E, and 
D in Equation (26) means the NED coordinate with cylindrical corrections. 
$\beta_{\left\{\theta ,r,D\right\}}$ are the biases of the radial, angular, and altitude terms, respectively. 
$v_{\beta_{\theta }}$ components and 
θ mean new position and new angular position after 
$\beta_{\theta }$ rotation correction.

Figure 13 shows the error distributions of the raw position estimation and error distribution after applying the correction Equation (26) with the parameters of Table 1.

For each drawing in Figure 13, the distributions of the real data are shown in blue and with a blue line their error distribution function [51]. The orange line shows a Gaussian distribution equivalent to the real data in Equation (27). For the NED coordinate, three different figures are represented: raw data Figure 13a–c, data after cylindrical correction using the mean of the raw data Figure 13d–f, and data after cylindrical correction according to the median of the raw data Figure 13g–i. In addition, the mean value and the standard deviation of each case represented by 
μ and 
σ, respectively, are indicated on each graph in their legend.
(27)
$$N\left(\mu ,\sigma ,x\right)=\frac{1}{\sigma \sqrt{2\pi }}e^{-\frac{{\left(x-\mu \right)}^{2}}{2\sigma^{2}}}$$

Figure 13 shows how correction Equation (25) modifies the structure of the error distribution for the cases of 
N and 
E coordinates, when comparing the blue with the orange lines. It can be seen in Figure 13a,b how the initial distribution is close to a Gaussian distribution Equation (27) and Figure 13d–h.

The effect on the Gaussian approximations in mean and standard deviation represented in Figure 13 is quantified in Table 4.

Table 4 shows the effect of mean and standard deviation on the data when applying the correction Equation (25) using the mean and median bias value indicated in Table 3.

The standard deviation rows of Table 4 decrease one order of magnitude in all coordinates when applying the correction Equation (25). The same effect can be seen in Table 5 when the RMSE Equation (28) of each NED coordinate is calculated. This metric can be considered as an indicator of accuracy.
(28)
$$RMSE=\sqrt{\frac{1}{N}\sum_{i=1}^{N}{\left({\hat{x}}_{i}^{n}-x_{GT_{i}}^{n}\;\right)}^{2}}$$

### 4.3. Landing Evaluation

To evaluate the landing system, we propose to test twenty landing missions from the same position at (10, 10, −20) NED meters to the helipad. The twenty flights are divided into four groups corresponding to using the raw landing system without correction (
without) Equation (2), applying the bias correction Equation (25) (
Bias), with bias correction and a mean filter (Section 3.1.3) with a sliding time window (
Mean&Bias) and a median filter together with the bias correction (
Median&Bias). For each mission, we use as quality metrics the landing trajectory distance Equation (29), time to land Equation (30), and landing accuracy. In addition, the results are compared with classical linear descent setpoints with 
α=0.7 [m/s].
(29)
$$Dist=\sum_{k=1}^{K}\left\|p_{k+1}^{n}-p_{k}^{n}\right\|$$
(30)
$$Time=t_{start}-t_{land}$$
where the 
k term means temporal step starting in 
$t_{start}$ instant and ending in 
$t_{land}$ moment. 
$p_{k}^{n}$ represents the global position at 
k instant in the NED reference frame.

For the flights’ analysis, we use the information obtained from the logs recorded by the flight controller in each flight and unloaded with the ground control station (GCS). In particular, we focus on the global positioning of the UAV. This positioning is given by the EKF2 fusion system [52] integrated in PX4 and with the specific sensor parameters indicated in the SITL [33] (Section 4.1). To ensure that we evaluate exclusively the landing phase of the UAV, we study the trajectories from the instant 
$t_{start}$ where the height gradient is detected to be negative, and the altitude is less than 99.8% of the altitude desired. The final instant 
$t_{land}$ is obtained when the helipad is reached by the same method as 
$t_{start}$.

Finally, third quality metric, landing accuracy, is obtained with the RMSE of the last ten position samples of the NED coordinates (without altitude). In this way, we ensure that we are on the ground with the same value plus a precision error. For this, we take as ground truth the control position of the marker.

Figure 14 shows the behavior of the aircraft when activating the landing system with the four modes corresponding to the landing system without correction (Figure 14a), landing with bias correction (Figure 14b), landing with bias correction and mean filter (Figure 14c,d), and landing with bias correction and median filter. The five trajectories in each of the figures show a different flight, using the corresponding landing mode in each case. The total number of flights is five for each landing mode, i.e., twenty flights.

Figure 15 illustrates the temporal behavior of each analysis group, showing the three global position components: latitude, longitude, and relative altitude. In addition, our proposal is comparing with linear descent, paying special attention to altitude evolution Figure 15e,f. In this case, estimated altitude and setpoints are shown for the exponential proposal and linear descending.

It can be seen in all cases in Figure 15 how the position of the landing pad, defined by a blue dashed horizontal line, is obtained in the stationary state. The effect of the error in precision is shown with an oscillating behavior (blue dotted line). However, when bias Equation (26) and filtering (Section 3.1.3) corrections are applied, the effect is damped.

The linear decrease approach converges in latitude and longitude (Figure 15b,d), but the system finds it difficult to dampen the spin effect. In the case of an exponential decrease, the inter-waypoint spin effect is considerably less than linear (Figure 15a–d). In both cases, the linear and exponential altitude decreasing approaches (Figure 15) show when error correction is applied and filtering estimation inter-waypoint noise spin effect is smoothed.

Figure 15e,f show a small break at the end of the trajectory that exceeds the reference and picks it up again. To identify the instant to obtain the landing surface, we keep the first term that satisfies the gradient and proximity conditions explained above.

This effect may be a consequence of a decoupling of the fusion system in the estimation of the height, for example for giving more weight to the estimation than to the measurements in the EKF2 filter. Therefore, when identifying the instant of reaching the pad, we are left with the first term that meets the conditions of gradient and proximity explained above.

The total of twenty flights with exponential decreasing approach and the other twenty flights with linear decreasing approaches are summarized in Table 6 and Table 7. These tables are built with the mean and median values of all flights, so the quality metrics means the mean and median values of all flights.

Table 6 and Table 7 show the results of the exponential and linear decreasing approaches grouped for easy comparison. The previous tables show for all cases that the exponential descent proposal improves the results of the linear descent, emphasizing that in the best-case scenario of the linear approach (
Mena&Bias), the trajectory distance is reduced by 32%, the time to land by 61%, and the RMSE of the precision landing by 12%.

In both tables, the mode that provides the minimum mean landing trajectory distance is the bias correction together with the median filter. The minimum mean time to land is provided by the bias correction together with the median filter and the minimum mean RMSE is again provided by the bias correction together with the mean filter. Finally, the similarity between the mean and median values in Table 6 and Table 7 are a good indicator of normal distribution.

## 5. Conclusions

Through the study of the global position estimation system error of Section 4, an angular and radial bias was identified. In addition, it was shown how the error distribution increases its degree of dispersion with the distance to the origin. This position error was initially modeled in a cylindrical space in Equation (24) and transferred to the NED reference space under the transformation Equations (25) and (26). The corrections over cylindrical space produced a structural transformation of the position error distribution, approximating its distributions to the Gaussian error.

On the other hand, it was verified how using a system aimed at smoothing trajectories between waypoints can produce a spin effect if the new waypoints are updated with a frequency such that the UAV cannot obtain the previous target and these new waypoints correspond to the same position, but with high uncertainty. Therefore, the path planning system with path smoothing between waypoints can work as an error amplifier performing a circular trajectory.

Finally, we conclude that the combination of an exponential altitude decrease, together with the correction of systematic estimation error and a sliding time window filtering, improves all three quality metrics proposed and reduces the effect of the inter-waypoint noise spin effect. These results facilitate the development of new applications that require a lightweight but robust precision landing strategy.

## Figures and Tables

**Figure 1 sensors-22-03625-f001:**
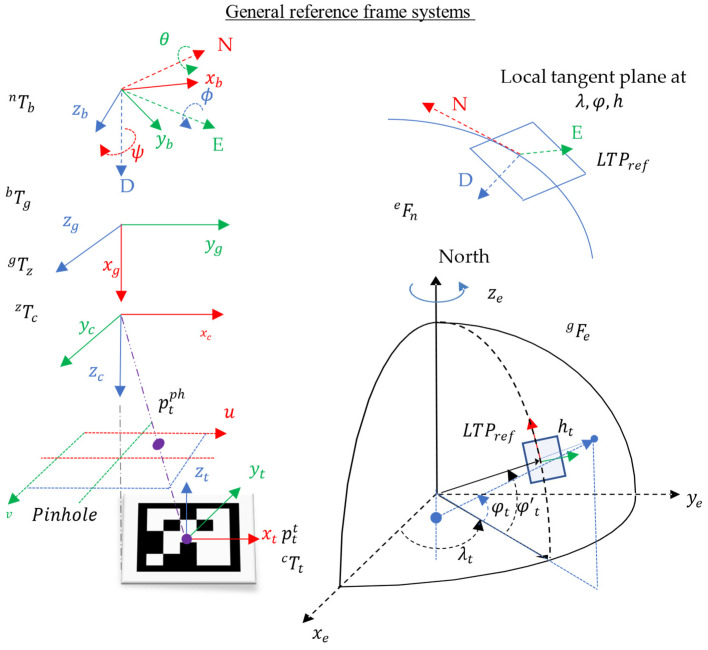
General reference frame systems. Reference frames bottom-left to up: helipad (target), pinhole camera model (image plane), camera, gimbal socket, body, NED. Reference frames right up to down: NED, ECEF, Global.

**Figure 2 sensors-22-03625-f002:**
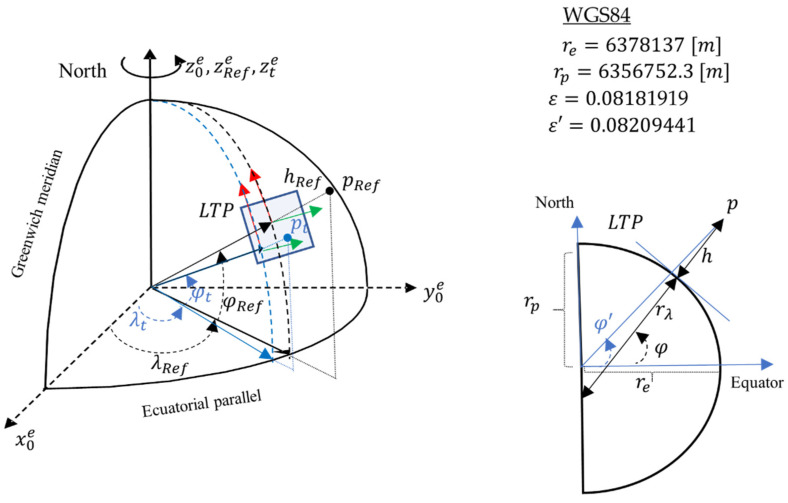
LTP, ECEF, and WGS84 reference systems and geometric relationships.

**Figure 3 sensors-22-03625-f003:**
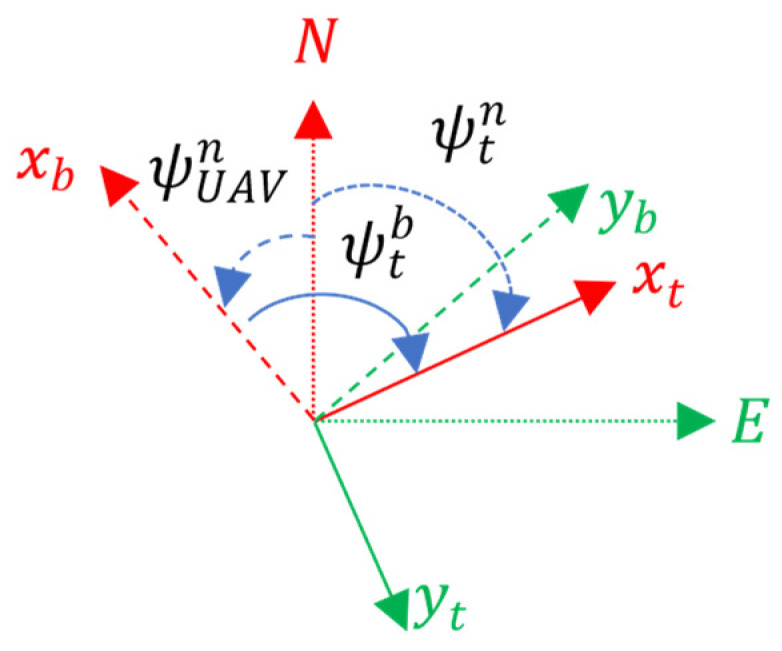
Helipad azimuth set formulation.

**Figure 4 sensors-22-03625-f004:**
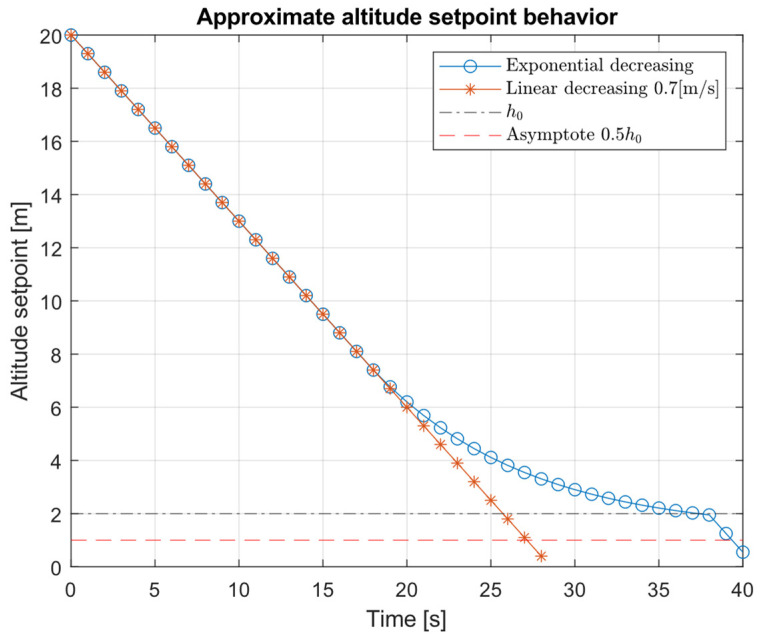
Approximate altitude setpoint evolution.

**Figure 5 sensors-22-03625-f005:**
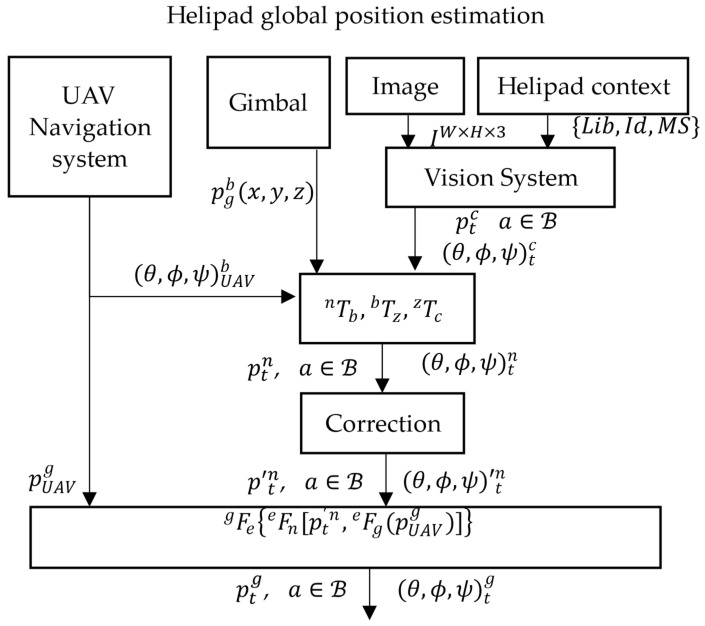
Helipad global position estimation system.

**Figure 6 sensors-22-03625-f006:**
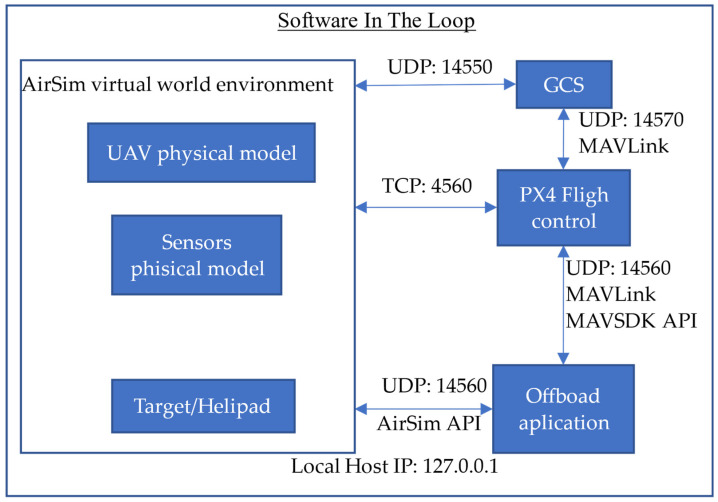
SITL Communication and protocol diagram.

**Figure 7 sensors-22-03625-f007:**
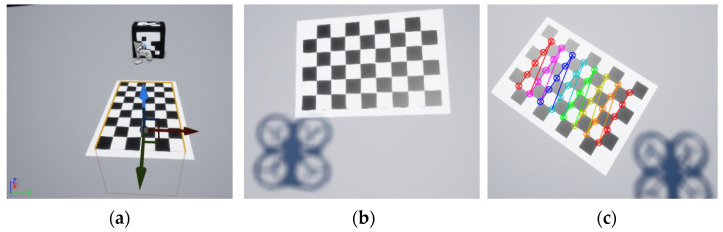
Simulation environment in the calibration process: (**a**) Image of the calibration pattern in the AirSim reference frame; (**b**) Random image of the image registration process; (**c**) Example of reprojection error.

**Figure 8 sensors-22-03625-f008:**
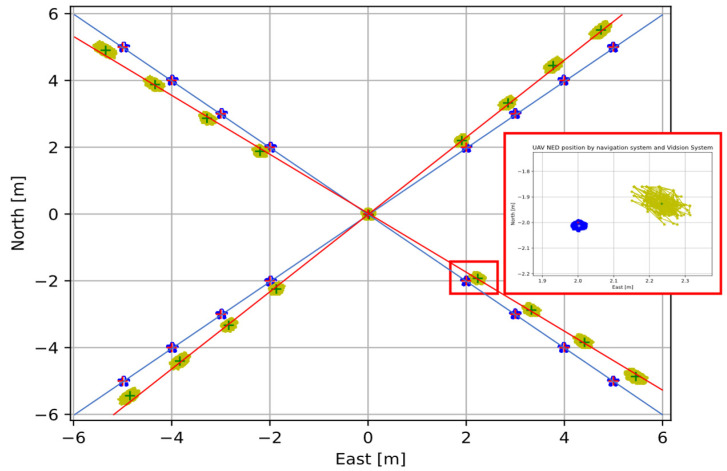
Data registration in twenty different flights. Yellow, UAV positions with vision system. Blue, UAV positions with navigation system. Blue and red line, linear approaches between navigation and vision system data centers, respectively. Red box, zoom in [−2, 2, 10] NED position.

**Figure 9 sensors-22-03625-f009:**
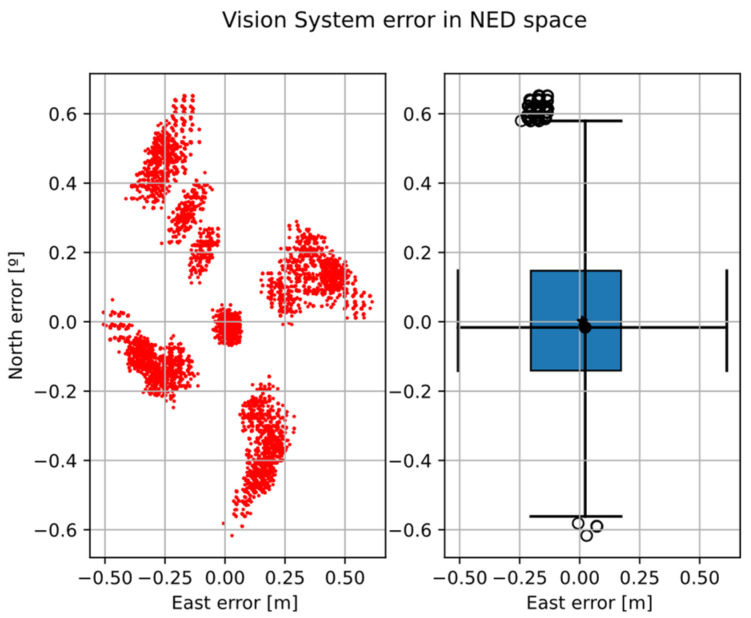
Vision system error in north-east plane coordinate. Left, scatter distribution; Right, north-east boxplot with 1.5 whis.

**Figure 10 sensors-22-03625-f010:**
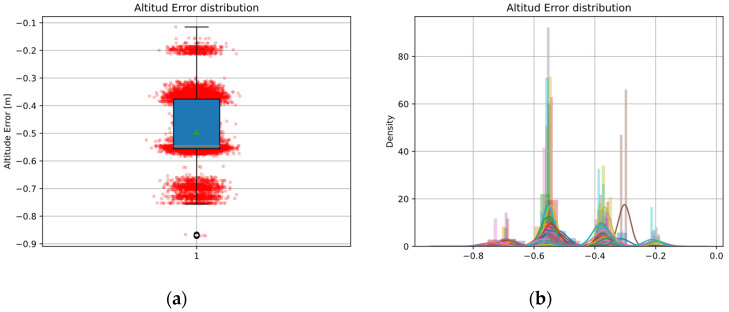
Altitude error: (**a**) Scatter and boxplot. Green triangle: mean, orange line: median; (**b**) Altitude error distribution for twenty different register positions.

**Figure 11 sensors-22-03625-f011:**
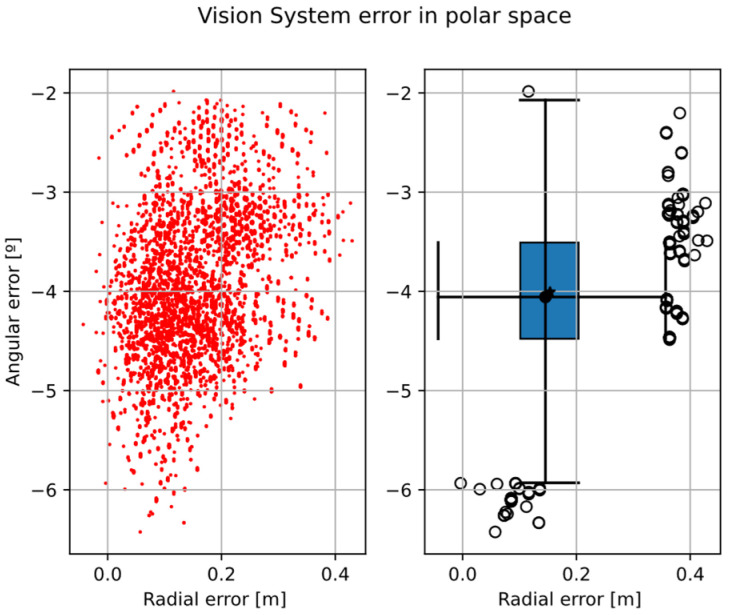
Vision system error in polar space. Left, scatter distribution; Right, 2D boxplot with 1.5 whis.

**Figure 12 sensors-22-03625-f012:**
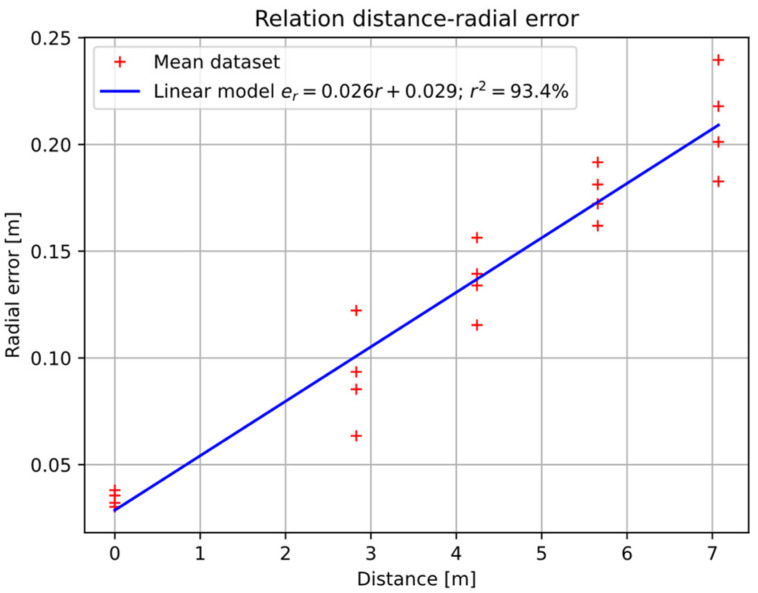
Relationship between distance and radial error. Red + symbol, centroid of each position. Blue line, linear model fitted by least squares.

**Figure 13 sensors-22-03625-f013:**
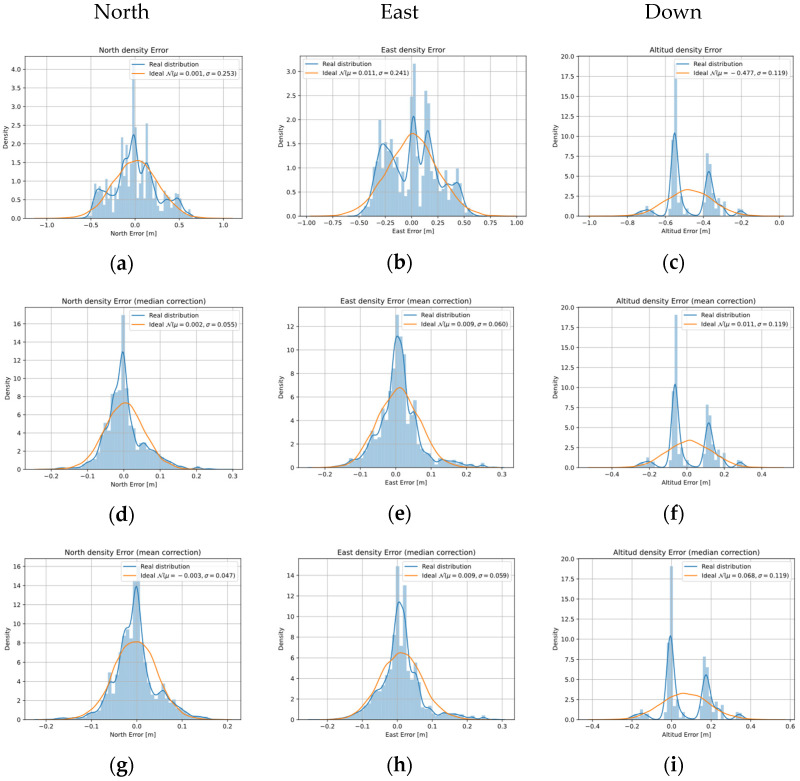
NED coordinates density error distribution. North-East-Down data without correction (**a**,**b**,**g**). Data with mean cylindrical correction (**c**,**d**,**h**). Data with median cylindrical correction (**e**,**f**,**i**).

**Figure 14 sensors-22-03625-f014:**
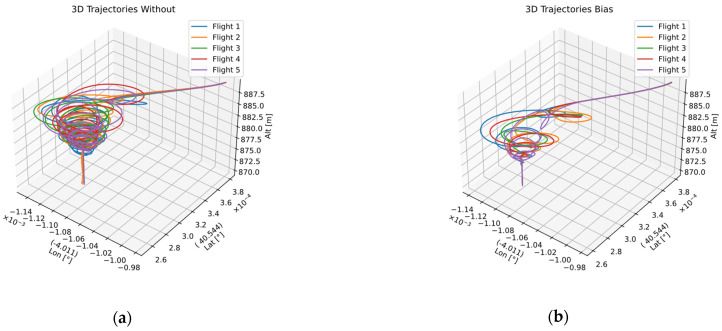
Five-flight 3D graphics for each of the four groups: (**a**) without correction; (**b**) bias correction; (**c**) bias and mean filter; (**d**) bias correction and median filter.

**Figure 15 sensors-22-03625-f015:**
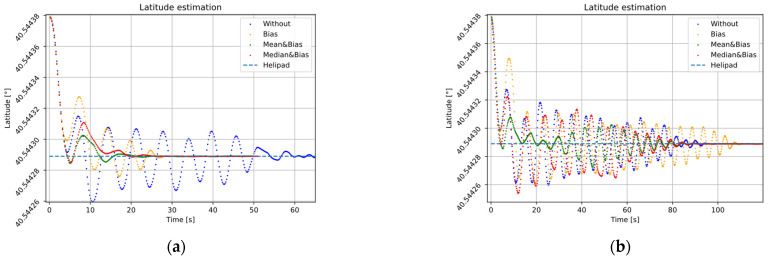
Time evolution of the latitude, longitude, and altitude of four flights with different correction modes in the landing phase: (**a**,**b**) latitude, (**c**,**d**) longitude, and (**e**,**f**) altitude. First column (**a**,**c**,**e**) exponential decrease, second column (**b**,**d**,**f**) linear decrease.

**Table 1 sensors-22-03625-t001:** Sensor parameters simulated in AirSim.

Sensor	Parameters
Barometer IMU GPS Magnetometer Distance	Default AirSim settings [49]
Gimbal-Camera	$\mathrm{Resolution}\;W\times H:\;640\;\times \;480\;\mathrm{Field}\;\mathrm{of}\;\mathrm{view}\;(\mathrm{FOV}):\;95\;\mathrm{Depth}\;\mathrm{of}\;\mathrm{field}\;\mathrm{focal}\;\mathrm{distance}:\;100\;\mathrm{Depth}\;\mathrm{of}\;\mathrm{field}\;\mathrm{focal}\;\mathrm{region}:\;100\;\mathrm{Depth}\;\mathrm{of}\;\mathrm{field}\;F-\mathrm{Stops}:\;2.8\;\mathrm{Target}\;\mathrm{gamma}:\;1.5\;\;p_{c}^{b}\equiv p_{g}^{b}=\left[0,0,0.1\right]\;{\left(\theta ,\phi ,\psi \right)}_{g}^{b}=\left(0,-\frac{\pi }{2},0\right)$

**Table 2 sensors-22-03625-t002:** Internal camera parameters.

Parameter	Value
$f_{x}$	293.35 [mm]
$f_{y}$	293.31 [mm]
$c_{x}$	319.64 [px]
$c_{y}$	239.64 [px]
$\mathrm{Distortion}\;\mathrm{coefficients}\;k_{i}$	$\left\{16.44,\;35.89,\;6.35,-6.35,\;100.7\right\}\times 10^{-4}$

**Table 3 sensors-22-03625-t003:** Stationary error.

	$\beta_{\theta }\;[{}^{\circ}]$	Distances [m]	$\mathrm{Altitude}\;\beta_{D}\;[m]$
Mean	−4.007078	0.153636	0.488412
Median	−4.056872	0.145540	0.545400
Var	0.512071	0.005182	0.014129

**Table 4 sensors-22-03625-t004:** Gaussian density distribution approximation.

	$N\left(\mu_{i},\sigma_{i}\right)$	Raw	Mean	Median
**North**	$\mu_{N}\;\left[\times 10^{-4}\right]$	7.930	20.200	17.120
	$\sigma_{N}\;\left[\times 10^{-2}\right]$	25.334	5.293	5.454
**East**	$\mu_{E}\;\left[\times 10^{-2}\right]$	1.087	0.851	0.882
	$\sigma_{E}\;\left[\times 10^{-1}\right]$	2.407	0.595	0.594
**Altitude**	$\mu_{D}\;\left[\times 10^{-1}\right]$	−4.774	0.110	0.680
	$\sigma_{D}\;\left[\times 10^{-1}\right]$	1.191	0.119	0.119

**Table 5 sensors-22-03625-t005:** RMSE value.

Coor./RMSE	Raw	Mean	Median
** $\mathrm{North}\;\left[\times 10^{-2}\right]$ **	6.418	0.280	0.298
** $\mathrm{East}\;\left[\times 10^{-2}\right]$ **	5.804	0.361	0.361
** $\mathrm{Altitude}\;\left[\times 10^{-2}\right]$ **	2.421	0.143	0.188

**Table 6 sensors-22-03625-t006:** Mean value of quality metrics.

Exp.|Linear	Distance [m]	Time [s]	$\mathrm{RMSE}\;\mathrm{Landing}\times 10^{-7}$
**Without**	131.25|205.88	143.77|136.37	1.650|1.74
**Bias**	57.07|192.15	44.30|131.62	1.011|2.38
**Mean&Bias**	37.44|54.73	44.64|116.09	0.875|1.17
**Median&Bias**	37.91|67.25	43.62|131.07	1.119|2.68

**Table 7 sensors-22-03625-t007:** Median value of quality metrics.

Exp.|Linear	Distance [m]	Time [s]	$\mathrm{RMSE}\;\mathrm{Landing}\times 10^{-7}$
**Without**	130.14|205.95	144.93|136.37	2.885|2.12
**Bias**	57.25|195.46	44.54|131.81	1.193|2.43
**Mean&Bias**	34.89|60.68	44.95|116.09	1.037|1.18
**Median&Bias**	36.21|85.58	44.34|132.73	1.042|2.70

## Data Availability

Not applicable.

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
