# Peer review of "Error Reduction in Vision-Based Multirotor Landing System"

_sensors, 2022, doi:10.3390/s22103625_

Round 1

Reviewer 1 Report

This paper proposes a vision-based multirotor landing system. A position error correction module is designed to increase the landing precision. The proposed system is evaluated in a simulation environment and improvement over raw landing system has been demonstrated. 

  1. However, it is not clear the novelty of the proposed method. As the authors say, there already exist many vision-based landing systems.  However, there is no discussion of the difference between the proposed method and existing work. For the evaluation part, only results are presented by disabling and enabling the proposed error correction module with the raw landing system which is also designed by the authors. Such evaluation process is essentially an abalation study only. Please add the evaluation between the proposed method and some existing baseline methods to truly reflect the novelty and advancement of the proposed method.
  2. The writing of the paper needs to be refined significantly:
    1. For the title, it is better to use “vision-based multirotor landing system” instead of “multirotor landing vision-based system”.
    2. A section named related work is missing. It is important to include the discussion of the state-of-the-arts, at least vision based multirotor landing system and the difference between the proposed work with the existing works. Please add such discussion.
    3. What is the context information from the helipad? What is the difference between this information and the information extracted from vision system? In line 278,  it mentions that “The helipad global position estimation system is responsible to integrate the vision 278 system, the heliport context information, the gimbal, and the UAV navigation states, to 279 provide the landing strategy the helipad global position.” But in Figure 5, the input from heliport context information is missing.
    4. Many sentences are written with grammar issues: to name a few, “. In Alvika Gautam et al. [1] review 34 of autonomous landing, describe the relationship between sensorization by the navigation …” in line 34-35. “the results of the study show how estimation error analysis and filtering of 69 the estimates with sliding time window filters,” in Line 69-70.
    5. A lot of typos: to name a few,  “specifically where or how is the helipad where you want to land” in Line 31-32; “a precision landing system” in Line 63.

Author Response

Dear member of the reviewing Sensors-MDPI committee,
First, we would like to thank the reviewer for his/her comments. His/her comments and suggestions were very helpful, and we believe that have helped to improve the manuscript “Error reduction in multirotor landing vision-based system”.

We attach a document with the answers to your comments.

Thank you again for taking the time and effort to review our work.

Kind regards,
Principal author, on behalf of the authors.

Reviewer 2 Report

In this paper, the authors propose a landing system based on monocular vision and navigation information to estimate the global position of a helipad. In doing so, the global position estimation system includes a position error correction module by transforming the cylinder space and a filtering system with a sliding window. In conclusion, it can be seen that the landing system is evaluated with 3 quality indicators showing how the proposed correction system together with the stationary filtering improves the raw landing system.
The topic of the article itself is quite interesting and relevant. Therefore, the article should be of interest to the readers of the journal. 
I have the following comments on the article:

1) The introduction is written rather weakly, as the key differences between this study and other works are not emphasized. It would have been desirable to highlight the unique features of the study in the article more vividly. 

2) It seems that not all symbols are defined in equations 1 and 2. Their description is not quite clear.

3)It is not quite clear why this function was chosen in (17). What is its advantage?

4) In their conclusions, the authors refer to section (3.1.3), but it looks as if they are referring to an equation. 
Also, the conclusions look like an enumeration of some subtasks without a global conclusion and generalization that would show the value of the study as a whole.

Author Response

(The authors gave the same response as above.)

Reviewer 3 Report

Please find attatched detailed review in the PDF file.

Author Response

(The authors gave the same response as above.)

Round 2

Reviewer 1 Report

The revised version has addressed my comments properly.

Reviewer 2 Report

The authors have revised the manuscript quite well, and the authors have responded to my comments. I have no further questions.

Reviewer 3 Report

All suggestions have been taken into consideration. The article has been substantially modified. The experiment has been improved and the results are clear. Thank you.